# Policy, Institutions and Regulation in Stormwater Management: A Hybrid Literature Review

**Carlos Novaes** [1,*] and **Rui Cunha Marques** [2]

1   CERIS, Instituto Superior Técnico (IST), University of Lisbon, Av. Rovisco Pais, 1049-001 Lisbon, Portugal
2   RCM2+ Research Centre for Asset Management and Systems Engineering, Lusófona University, Campo Grande, 376, 1749-024 Lisboa, Portugal; rui.marques@ulusofona.pt
*   Correspondence: cnovaes.augusto@gmail.com

**Abstract:** Policies, Institutions and Regulation (PIR) aspects matter for different sectors' growth and inclusive sustainable development, but there is little information in the literature on how to evaluate the effects of PIR on management options and outcomes or, on how positive results PIR changes can bring. In terms of stormwater management systems, or urban drainage, PIR is also a controversial and absent matter. Multidisciplinarity, several actors, countless formal and informal rules, and strong contextual path dependence make the subject complex and intricate. Considering the enabling environment, an alignment between policies, institutions and regulations is required to achieve good results and provide sustainable services. This study conducted a hybrid literature review of peer-reviewed papers in this field to provide an overview of how researchers have been studying PIR relations. The gaps show that the understanding of the PIR is fragile, as an important element for analyzing of results to be achieved, including SDG6, the financing and obtaining funds, guarantees and grants for the execution, delivery, operation and maintenance urban stormwater services and infrastructure. The contribution of this review is not only about what exists, but also mainly about what does not exist, since the void keeps waiting to be filled.

**Keywords:** drainage services; institutions; literature review; policy; regulation; stormwater management





## 1. Introduction

Policies are the result of the perceptions of public decision-makers (agents of the executive and legislative branches of government) and private (directors and managers) about the (not always) real problems of society. Institutions are the formal (laws, decrees, and regulations) and informal (habits and customs) rules of the game, and regulations are responsible for enforcing the rules.

Thus, policies provide the guidelines, institutions provide the rules to be followed, and regulations control actions at all stages, from planning to execution. The incentives for actors and organizations on each side of the tripod—PIR—may be different, but they must be aligned for actions to materialize

The issue is how to establish incentives that will create a continuous motivation to make a reality in terms of access, quality of service, financial resources, and environmental well-being [1].

This research aims to observe how the subject involving PIR, regarding the management of urban stormwater, has been addressed by the literature. Analyzing the available publications since the 1980s, when these matters started to be investigated, allows us to identify the trends, best practices and the gaps in the literature, particularly the existing themes and approaches and those that are absent, requiring research and deepening of discussions in technical and academic circles. For this purpose, a hybrid literature review methodology was adopted. It was structured into three phases, allowing a broad analysis of the evolution of the subject in the literature, over recent decades, a period in which

there were major changes in urban environments, population growth, climate change and aging infrastructures, generating pressures on systems and infrastructure, whose response is not always considered satisfactory in terms of the performance [2]. In the first phase, through a quantitative survey of the articles, the areas and locales with production were identified, which may be of interest to the academy, by the similarity between problems, even though a set of papers shows a great distribution of the topics of research. Therefore, a second phase was carried out to refine the articles found in the first phase (elimination process) and to match them with the aims of the research. Finally, a third phase related to the detailed analysis of the articles was developed (analysis process). This phase comprises a quantitative, semantic, and narrative analysis, which brings with it the identification of the main themes and their connections, complemented by a conceptual map generated from these themes.

To the best of our knowledge, this is the first literature review article on the PIR and managerial issues of stormwater management systems. Moreover, the contributions of this article are also based especially on communicating an essential perspective to scholars and decision makers as to the importance of the subject PIR, besides the gaps in the literature, and its importance also for the development of management necessary actions.

This paper comprises five sections. After this introduction, Section 2 describes the research methodology, Section 3 presents the results, discussed in Section 4 with the main research areas, and Section 5 draws conclusions and suggests future research.

## 2. Research Methodology

A literature review can provide information on the state of the art of a certain subject, contributing to those who are interested in the themes by starting from what is already known or even from what is unknown, but which requires more attention at certain times. This is the case for stormwater management, which requires reflection and exchange of experiences, due to its complex transdisciplinary nature and the absence, in many cities, of a clear institutionalization of services and establishment of responsibilities [3]. Even with similar regulatory and institutional frameworks, there are places where stormwater management systems have good results and others where this is not the case.

In the absence or existence of little systematized information about management and especially about the incentives provided by the alignment of PIR, we took the challenge of carrying out a hybrid literature review (HLR), a combined method comprising the narrative and systematic quantitative review methods, supplemented by semantic network analysis (Word Cloud and Concept Mapping) that could bring to light what exists and illuminate paths to follow about what does not exist and that, therefore, needs to be completed through studies, analysis, and reflections that can contribute to elucidating issues and pointing out solutions to urban stormwater management systems with a focus on the PIR approach. HLR methodology was adopted following three phases: search, elimination and analysis, resented in Figure 1.

Figure 1 describes the structure of the search, and the number of documents selected, while Table 1 presents the keywords and quantity of obtained documents. The review is based on the adapted Preferred Reporting Items for Systematic Reviews and Meta-Analyses for Protocols (PRISMA-P) guidelines.

In stage 1, the search engine Scopus was used, to retrieve paper articles and allows a subject's overview.

The keywords related to this study were grouped: institutions, policy, regulation, drainage, stormwater, and rainwater.

Stage 2 considers the elimination of duplicated and non-relevant papers, the former by the automated Scopus combination process and the latter by a visual examination, resulting in 112 papers selected. Stage 3, process analysis, contains a systematic quantitative review, semantic, and narrative analysis.

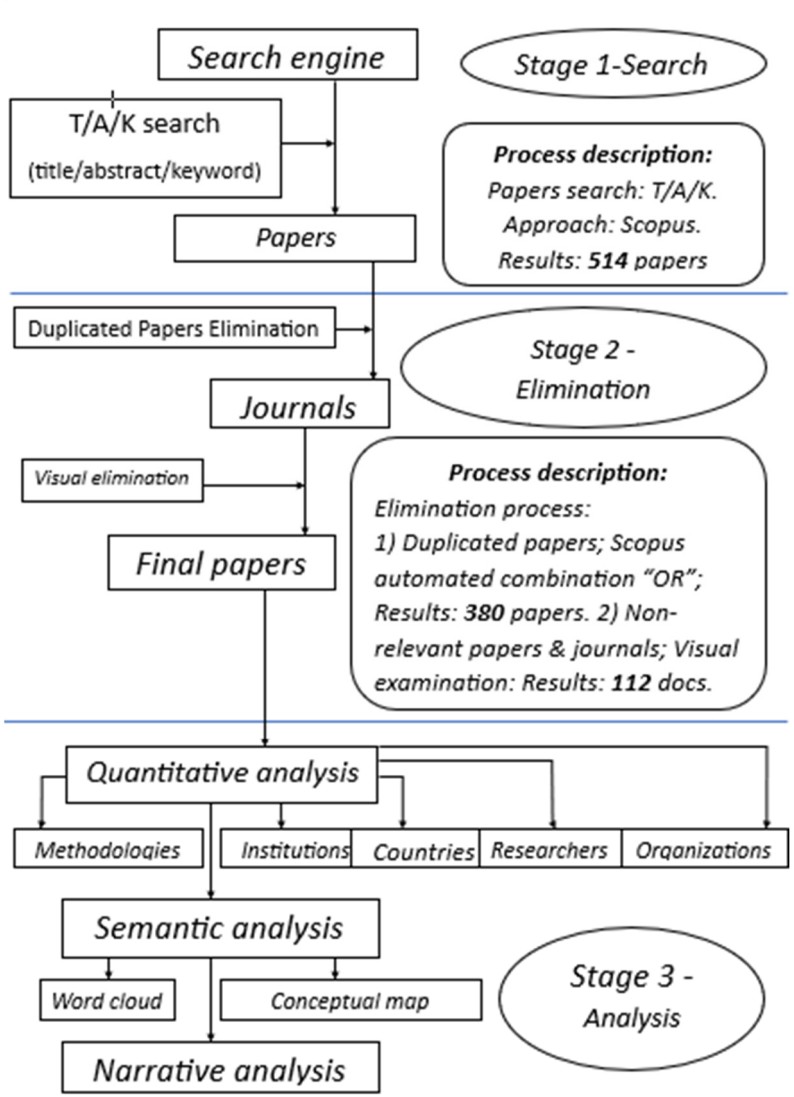

**Figure 1.** Research framework.

**Table 1.** Results of stage 1 of the search process through the Scopus database.

| Search Number | Keywords | Results |
|---|---|---|
| 1 | Poli* Institut*Regul* Drainage | 24 |
| 2 | Poli* Institut*Regul* Stormwater | 17 |
| 3 | Poli* Institut*Regul* Rainwater | 9 |
| 4 | Poli* Regul* Rainwater | 41 |
| 5 | Poli* Regul* Stormwater | 115 |
| 6 | Poli* Regul* Drainage | 183 |
| 7 | Institut*Regul*Drainage | 62 |
| 8 | Institut*Regul*Stormwater | 39 |
| 9 | Institut*Regul*Rainwater | 24 |
| Total | | 514 |

Note: The use of *(wildcards) in Scopus allows to the recovery word variations (e.g., "Regulation" and "Regulatory").

## 3. Results

### 3.1. Systematic Quantitative Review

#### 3.1.1. Publications Distribution by Time

The first stage of the quantitative systematic review was the analysis of the number of papers published. The distribution of the 112 documents, researched over a period between the 1980s and 2021, is presented in Figure 2. There has been a growing trend, which allows us to assume that there is an increasing interest in the subject.

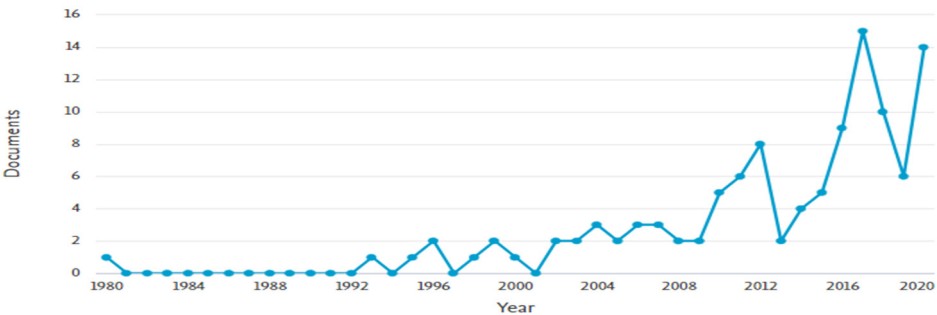

**Figure 2.** Documents distributed by year of publication.

#### 3.1.2. Publications by Subject Area and Journals

The highest number of papers area is Environmental Science with 97, or 41.8% followed by Social Sciences with 49, or 21.1%, and Engineering with 25 or 10.8% of the total. Some of them are associated with different subject topics. The top ten journals with the most articles or more than three articles are shown in Table 2 below. *The Journal of Environmental Management* and *Water* journals are the ones with the most articles on the subject. In total, 73 journals published articles.

**Table 2.** Number of papers by journal.

| Journal | Q Index | n° of Papers |
| --- | --- | --- |
| Journal of Environmental Management | Q1 | 6 |
| Water Switzerland | Q1 | 6 |
| Urban Water Journal | Q2 | 4 |
| Journal of Cleaner Production | Q1 | 4 |
| Journal of Water Resources Planning and Management | Q1 | 4 |
| Water Science and Technology | Q3 | 4 |
| Sustainability Switzerland | - | 3 |
| Journal of Hydrology | Q1 | 3 |
| Journal of The American Water Resources Association | Q1 | 3 |
| Landscape and Urban Planning | Q1 | 3 |
| Others (2 papers each journal) | N/A | 18 |
| Others (1 paper for each journal) | N/A | 54 |
| Total | | 112 |

#### 3.1.3. Publications by Authors, Organizations, and Countries

There is more than one author in many studies, which totals more than three hundred authors in the 112 documents or an average of more than 2.0 authors for each study in 73 journals. The most published author is J.B. Ellis from the Natural Environment Research Council in the UK with four publications, as shown in Table 3.

**Table 3.** Number of published studies by author and author's score.

| Authors | Studies | Author Scores |
|---|---|---|
| Ellis, J.B. | 4 | 2.09 |
| Brown, R.R. | 3 | 1.00 |
| Schimtt, T.G. | 3 | 1.94 |
| Butler, D. | 2 | 0.25 |
| Campisano, A. | 2 | 0.52 |
| Ciaponi, C. | 2 | 0.63 |
| Cook, S. | 2 | 0.38 |
| Deletic, A. | 2 | 0.24 |
| Duke, L.D. | 2 | 1.00 |
| Ettrich, N. | 2 | 0.42 |
| Hogue, T.S. | 2 | 0.47 |
| Iftekhar, M.S. | 2 | 0.36 |
| Kim, J.H. | 2 | 0.58 |
| Maksimovic, C. | 2 | 0.33 |
| Marsalek, J. | 2 | 0.51 |
| Matsler, A.M. | 2 | 0.68 |
| Papiri, S. | 2 | 0.64 |
| Petrucci, G. | 2 | 0.89 |
| Sharma, A.K. | 2 | 0.76 |
| Thomas, M. | 2 | 0.64 |
| Tjandraatmadja, G. | 2 | 0.25 |
| Todeschini, S. | 2 | 0.94 |
| Wong, T. | 2 | 0.48 |
| Zhu, D. | 2 | 0.72 |
| Others 136 authors | 136 | |
| Total | 188 | |

Concerning the main author's affiliation, Table 4 shows that 32.7% (64) of the total (196) authors are from twenty-eight organizations, most of all universities, and in twelve countries (Australia, Canada, China, Colombia, Germany, Israel, Italy, New Zealand, South Africa, Sweden, the UK, and the USA), and nearly 25% of these 64 studies (16) were produced in Australia and 23.4% in the USA (15). Table 5 illustrates the author's score calculated according to the formula [4] presented below and Table 6. The number of publications by country is shown in Table 7.

**Table 4.** Publication affiliations and authors' countries.

| Author's Affiliation | Country | Studies |
|---|---|---|
| Monash University | Australia | 6 |
| The University of Auckland | New Zealand | 2 |
| Seoul National University | South Korea | 2 |
| Middlesex University | UK | 3 |

**Table 4.** *Cont.*

| Author's Affiliation | Country | Studies |
|---|---|---|
| University of Melbourne | Australia | 3 |
| University of California, Los Angeles | USA | 3 |
| School of Ecosystem and Forest Science | Australia | 3 |
| Environment Canada | Canada | 2 |
| The University of Sheffield | UK | 2 |
| Arizona State University | USA | 2 |
| Purdue University | USA | 2 |
| Technische Universität Kaiserslautern | Germany | 2 |
| Università degli Studi di Catania | Italy | 2 |
| University of Oregon | USA | 2 |
| Colorado School of Mines | USA | 2 |
| Imperial College London | UK | 2 |
| Università degli Studi di Pavia | Italy | 2 |
| Institute of Ecosystem Studies | USA | 2 |
| Western Sydney University | Australia | 2 |
| Helmholtz Zentrum für Umweltforschung | Germany | 2 |
| Chinese Academy of Sciences | China | 2 |
| Fraunhofer Institut Industrial Mathematics ITWM | Germany | 2 |
| Texas A&M University | USA | 2 |
| Technion—Israel Institute of Technology | Israel | 2 |
| Shangai Jiao Tong University | China | 2 |
| CSIRO Land and Water | Australia | 2 |
| University of Exeter | UK | 2 |
| Fundación Universidad de América | Colombia | 2 |
| Others (1 each one) | N/A | 132 |
| Total | | 196 |

**Table 5.** Score distribution of countries, organizations (O), researchers (R), and papers (P).

| Country | O. | R. | P. | Score | Country | O. | R. | P. | Score |
|---|---|---|---|---|---|---|---|---|---|
| USA | 61 | 104 | 45 | 41.88 | Belgium | 1 | 3 | 1 | 1.00 |
| Australia | 23 | 50 | 15 | 12.87 | Poland | 1 | 2 | 1 | 1.00 |
| UK | 19 | 33 | 14 | 11.38 | Israel | 1 | 4 | 1 | 1.00 |
| China | 17 | 33 | 8 | 6.45 | Indonesia | 1 | 4 | 1 | 1.00 |
| Germany | 10 | 13 | 7 | 6.28 | Bangladesh | 3 | 3 | 1 | 1.00 |
| Canada | 10 | 15 | 6 | 5.22 | Tanzania | 1 | 1 | 1 | 1.00 |
| Italy | 9 | 20 | 5 | 5.00 | India | 1 | 2 | 1 | 1.00 |
| Brazil | 6 | 13 | 6 | 4.54 | Mexico | 1 | 2 | 1 | 0.46 |
| Colombia | 5 | 9 | 3 | 4.00 | Malaysia | 1 | 3 | 1 | 0.36 |
| N. Zealand | 2 | 11 | 4 | 3.64 | Greece | 1 | 1 | 1 | 0.32 |
| South Korea | 6 | 7 | 6 | 3.32 | Saudi Arabia | 1 | 1 | 1 | 0.18 |
| Spain | 5 | 6 | 2 | 2.00 | Sweden | 1 | 1 | 1 | 0.12 |
| France | 2 | 7 | 2 | 2.00 | S. Africa | 1 | 1 | 1 | 0.03 |
| Finland | 2 | 3 | 2 | 1.42 | Japan | 1 | 1 | 1 | 0.03 |
| Switzerland | 2 | 3 | 2 | 1.42 | | | | | |

**Table 6.** Score matrix for multiauthor papers (elaborated by authors based on [4]).

| Number of Authors | Order of Specific Author | | | | | |
|---|---|---|---|---|---|---|
| | 1 | 2 | 3 | 4 | 5 | 6 |
| 1 | 1.00 | | | | | |
| 2 | 0.60 | 0.40 | | | | |
| 3 | 0.47 | 0.32 | 0.21 | | | |
| 4 | 0.42 | 0.28 | 0.18 | 0.12 | | |
| 5 | 0.38 | 0.26 | 0.17 | 0.11 | 0.08 | |
| 6 | 0.37 | 0.24 | 0.16 | 0.11 | 0.07 | 0.05 |

**Table 7.** The main countries where the documents are published.

| Country | Documents Published |
|---|---|
| United States | 43 |
| Australia | 16 |
| United Kingdom | 14 |
| China | 8 |
| Brazil | 6 |
| Germany | 6 |
| Italy | 6 |
| South Korea | 6 |
| Canada | 5 |
| Colombia | 4 |
| New Zealand | 4 |
| Sweden | 3 |
| Finland | 2 |
| France | 2 |
| Israel | 2 |
| Spain | 2 |
| Switzerland | 2 |
| Others (1 each one) | 24 |
| Total | 155 |

### 3.1.4. Distribution by Countries 'Organizations' Researchers

Several published documents were higher than the number of studies (112) because one paper could be published in more than one country. There were 29 countries and 160 organizations, most of all universities, showing the subject's interest all over the world.

The contribution score of each country is shown in Table 5 using the multi-author paper score matrix of Table 6. The countries that contributed to the subject of the research during the period studied are identified in Tables 4 and 5, the first with the author's affiliations and, in Table 5, together with the number of institutions/universities, researchers, and papers. The USA, Australia, the UK, China, and Germany were the top five countries. In the US, 104 researchers from 61 organizations published 45 papers on the research subject, whereas in Australia, 50 scholars from 23 organizations contributed 15 publications during the period covered by this study. These results are reasonable because the US and Australia are internationally known as forerunners of the implementation of Stormwater Utilities [5]. Australia has also made good progress in institutional sectoral arrangements, which could be credited to the fact that the country has established a well-organized sanitation sector [6]. In Germany, 13 researchers from ten research centers published seven papers. Some developing countries, such as India and South Africa, despite having low

scores (1.00 and 0.03), published papers showing a growing interest in the subject. Table 3 above shows the twenty-four most active researchers with two or more papers, and the other 136 are all authors of just one paper.

Although contributing to most of the publications, nations such as the USA and Australia, have individual researchers with low scores because of the large number of researcher collaborations in particular publications. Identifying the contributions of countries, organizations, universities and authors is important for researchers and practitioners for future collaborations.

To assess the contributions of countries and researchers to this research, a formula proposed by Howard et al. [4] was used. This formula has been extensively used in previous review studies [7,8].

The formula is

$$\text{Score} = 1.5^{n-i} / \sum_i^n 1.5^{n-i}$$

where n = number of authors and i = order of a specific author. Applying this formula, each paper was assigned a score of 1.00. Based on the position of authors on a multi-author paper, the formula gives scores for authors. This formula assumes that a first author has made a greater contribution than a second author, a second author has contributed more than a third author, and so on, for example, if an author is the first author of Paper A, the second author of Paper B, and the third author of Paper C, assuming each paper which could also be used to score the author's country has three authors, the score of this author is 0.47 + 0.32 + 0.21 = 1. According to this methodology, each author's score was calculated, and after the duplicated authors were excluded, all were added to determine the contribution score of each country. We can conclude that both developed and developing countries are present, suggesting that this topic is of global interest.

### *3.2. Semantic Analysis*

#### 3.2.1. Word Cloud

Word cloud generators can be seen as exploratory tools and help find, in all titles, abstracts, and keywords, of the 112 documents, the word frequency of related terms enabling the differentiation of the high-frequency words, and the result is a visual word frequency cloud, like in Figure 3, with the 50 most frequent words. The software Version v3.6 (Word Cloud Generator by Monkey Learn) permits not just search by words (e.g., management, stormwater, regulations, and drainage) but related joint associated terms in expressions like "stormwater management" or "rainwater harvesting". Then, the software analyzes the words in two ways according to: the frequency of terms or according to their relevance. In short, the terms "regulation", "urban water management", "rainwater harvesting", "SUDS", "urban drainage systems", "stormwater management", "green infrastructure" (GI), "low impact development-LID" and "urban green space" are popular topics.

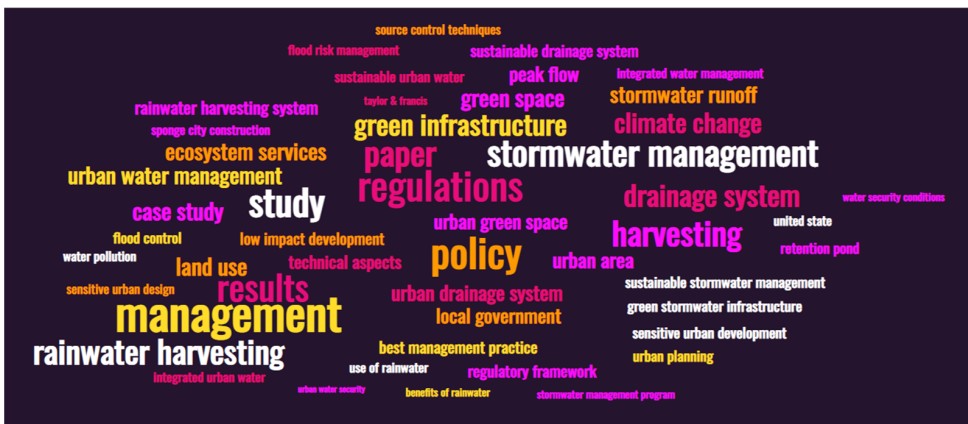

**Figure 3.** Research word clouds with the 50 most frequent terms.

3.2.2. Conceptual Map

A data mining tool can be used to analyze the content of textual documents and visually display information through a conceptual map representing the main themes and concepts and how they are correlated. It is possible to visualize both quantifying the importance of the main concepts, discussed in the document set and displaying the conceptual structure linking concepts. The tool used in this article—Leximancer 5.0—automatically extracts the most important concepts and builds a concept map. Three main sources of information about the document set are provided by the software: (a) the main concepts and their relative importance; (b) the strengths of links between concepts (how often they co-occur); and (c) the similarities in the context in which they occur.

Leximancer's results, in Figure 4, revealed seven themes, and the most cited inter-related themes were water, management, land, use, harvesting, implementation, and institutional. Leximancer concepts are defined as collections of words that occur together throughout the text. The thirteen main themes are water, management, stormwater, urban, harvesting, rainwater, systems, drainage, policy, use, development, implementation, and regulation. For example, the water theme shows a correlation with the concepts of cities, quality, urban, system, green, flood, infrastructure, and sustainable, and the management theme is associated with the concepts of policy, regulatory, integrated, research, local, development, stormwater, infrastructure, and study. The implementation theme is related to sustainability, urban, drainage, flood, infrastructure, and planning; use correlates with runoff, quality, control, and policies.

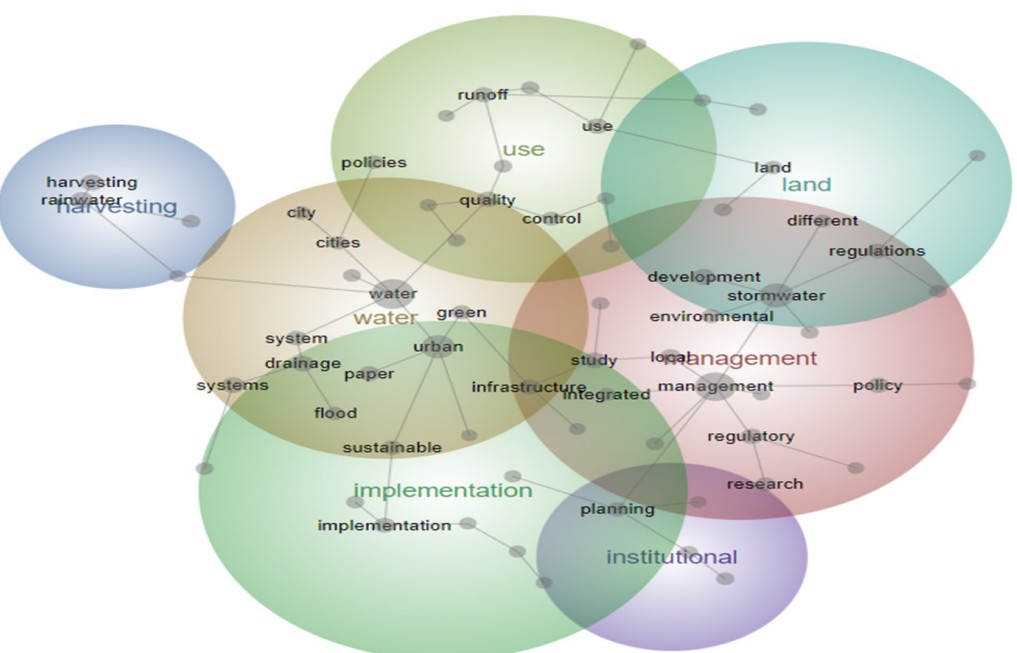

**Figure 4.** Conceptual map by Leximancer.

*3.3. Narrative Analysis*

Narrative analysis refers to the presentation and exposition, in more detail, of the main themes pointed out in the 112 studies obtained from the research carried out by the search engine Scopus.

The main issues and challenges in the urban stormwater management subject are related to the lack of institutionalization, policies, and of regulatory issues. According to a recent approach, systems performance depends on the alignment between the three elements of the PIR approach, which are absent or misaligned in most municipalities and jurisdictions [1].

According to the survey results, six areas of greatest interest, all interconnected, are found: institutional arrangements, policy implementation, public regulation, governance,

low-impact development (LID), and rainwater. Although interconnected, they can be grouped into three themes: institutions and governance, policy implementation, and regulation.

### 3.3.1. Theme 1: Institutions and Governance

There is a great concern, in terms of the sector organization, reflected in many articles (16 or 14%), related to institutions and governance of the water systems. In the articles referring to China [9] there is a context of hybrid coexistence in a "two-hand" system, i.e., on the one hand with power concentrated by the central government with investments in infrastructure and, on the other hand, with market principles and local governance powers. Another aspect highlighted in the Chinese context is economic development vs. water management and how to build a win–win strategy. Here, the sponge cities concept is emphasized [10]. Based on the idea of low-impact development (LID), the concept of Sponge Cities (SC) is broader, aiming to restore, through imitation, the original urban water cycle, i.e., the pre-development scenario. The aim is to seek the conditions of the basin regime prior to current development, with an emphasis on natural accumulation, infiltration and filtration or purification, but also on the reuse of run-off water, which is considered a resource, including to combat the phenomenon of urban heat islands (UHI). In addition to considering floods and droughts, the CS concept also considers the combination of the effects of new green infrastructure and existing drainage networks, as well as the replenishment of groundwater resources and the effects of the depth of the water table [11], as in Wuhan, a city with a high water table [12].

Regulate stormwater management in Germany, has two key tools: the compulsory connection and stormwater usage charges. The question is how these tools can link decentralization with the objectives of stormwater management and which institutional designs can integrate private owners in only one strategy not only for new areas but also for existing settlements [13].

Stormwater has increasingly become a resource [14] and not a flood or pollution problem, but change brings management risks and responsibilities [15]. The city of Salisbury in Australia manages flood risks and generates a profitable business from the storage of run-off water. This is a successful example of taking care of risks and seizing opportunities within the concept of Water Sensitive Urban Design/Low Impact Development WSUD/LID sponge city. Recycled stormwater, currently used for non-potable purposes, may soon, with advances in knowledge, be used for potable purposes. Industries such as Water-Sensitive SA, together with federal and local governments, have established an integrated system as a commercial business supplying recycled water for non-potable and industrial uses. With a small degree of treatment, the water from the wetlands and from storage by injection into aquifers can be used as a drinking water source. Australia developed a national water quality management strategy about 30 years ago and has produced guidelines, including manuals dealing with sewerage systems, specific effluents, like spilled fuels, fertilizers and pesticides, water recycling with health and environmental risk management, stormwater use and aquifer recharge management [16]. Each urban water alternative considered can have implications on how risk is conceptualized and managed, which involves how governance arrangements and risk allocation interact. Community perceptions are important, and different cities do not have the same perception of risk. In Chicago, in the USA, there are many aspects of convergence and divergence about this and broader stakeholder participation in urban governance and decision-making. Two dominant interventionist perspectives, in terms of stormwater management, interact and conflict: the infrastructural and the institutional. The first views laws and regulations and the second considers new rules and integrated management, with more robust economic instruments, as the best ways to improve stormwater management systems [17].

In Australia, policy changes are often recommended to promote the sustainability and resilience practices of the Murray-Darling Basin Authority (MDBA), to provide effective water regulation. A linear park project, River Torrens Linear Park (RTLP), a Green Infrastructure (GI), is an example of challenges in terms of institutional and political frame-

work and governance, because it involves many stakeholders and various political and institutional aspects, as presented in Table 8 [18].

**Table 8.** Factors influencing governance and the performance of the River Torrens Linear Park—a GI.

| Internal |
| --- |
| Project Design GOALS |
| Multiple objectives and drivers |
| Interpretation of goals by various actors |
| Upholding of the goals |
| Communication and Collaboration |
| Multiple stakeholders |
| State agencies and communities |
| Sharing of responsibilities among stakeholders |
| Maintenance and funding arrangements |
| Shared vision |
| External |
| Political and Institutional System |
| Leadership |
| Political restructuring |
| Financial strength |
| Framework for cross-scale engagement |
| Implementation of existing policies |
| Population growth |
| Legal Framework |
| Planning regulation |
| Development control and enforcement |
| Development new policies and guidelines |

The RTLP project is a good example of how the paradigm shift, from a traditional to the sustainable stormwater management system, can mean an increase in complexity, due to the entry of many actors. The more functions there are for stormwater, the greater the number of actors competing for decisions about the fate of this resource, often with distinct interests. In the case of the RTLP, Table 8, in a simplified way, tries to demonstrate many actors and aspects involved by showing explicitly the political, institutional, and regulatory systems. Schematically, it divides into internal and external factors, the latter being classified as PIR systems. The issue, in brief, means that the stormwater management paradigm shift brings multiple objectives and multiple actors, requiring not only a change in vision and mentality, but also a sharing of responsibilities.

A conventional drainage system considers rainwater to be a nuisance to cities and therefore believes that it should be removed from the urban environment as quickly as possible. A sustainable drainage system (SUDS), on the other hand, is based on the city's harmonious coexistence with stormwater, which should remain in cities for as long as possible and its use as a valuable resource for the well-being of the population is encouraged. Sustainable systems are better adapted to the environment and the people who live in it, for example by using the resource to combat water scarcity and the effect of urban heat islands (UHI) [19].

The implementation of Sustainable Drainage Systems (SUDS) has become a trend in the world, but in Poland, conventional stormwater systems continue to be the main methods of stormwater management, and the change depends on legal regulation financing barriers [19].

Australia's SUDS approach integrates urban planning and design with urban water cycle management, as the best management practice (BMP), together with Low-Impact

Development (LID), popular in the USA and Europe, but as in the UK [20] finds barriers in the limited awareness of planners, engineers [21] and others.

In Canada, one of the most decentralized countries of the Organisation for Economic Cooperation and Development (OECD), regarding governance, there is concern about the increasing complexity of basin management and water protection and how the relationship between decentralized and multilevel actors can hinder or help watershed management, especially the protection of water resources. In addition, the costs of stormwater management and maintenance are rising, and user pay implementation faces obstacles such as public perception and start-up costs [22].

Megacities' governance, a complex task, involves many aspects, such as the gap of investment funds to construct and maintain the water infrastructure in addition to an inadequate institutional framework, with deficiencies in legal and regulatory regimes [23]. In New York, the conditions under which the decision-makers operate must be understood by considering local problems, e.g., fragmented jurisdiction but unified stormwater management, under compliance with federal regulations, so that an adaptive governance approach is needed to manage this complex environment [24]. This institutional complexity, with many actors and new policy trends, such as rainwater harvesting [25], has impacts on stormwater governance [26], e.g., the institutionalization of rainwater harvesting through market-based tools [27], demanding government regulation, and GI projects [18].

### 3.3.2. Theme 2: Policy Implementation

The implementation of stormwater policies takes place according to the different characteristics and demands of each location but represents the stages of development of the issue in each one. For example, in Colombia, rainwater resource management has poor public policies integrating the urban design of cities [28] and also encounters barriers of inflexible policies and institutional fragmentation to make cities water-sensitive [29].

Decades of implementing rainwater detention systems, with little evaluation of their effectiveness, demonstrate how necessary it is to collect data and information, so that policies regarding these systems can be improved [30].

The growing interest in GI policies and their potential to provide stormwater management in conjunction with other urban systems, in many localities, such as Baltimore, represents a diverse portfolio, including not only detention basins but also filters, infiltration facilities, and swales, improving the water quality [31]. Despite being sustainable, GI implementation is slow, and gray infrastructure is usual in urban areas, showing differences between knowledge and practices, with many barriers from cognitive to socio-institutional arrangements [32].

In Italy, a simulation study, with different SUDS options, showed better performance by green roofs than permeable pavements, suggesting that innovative policies may encourage more private landowners to adopt this kind of GI [33].

For decades, in Germany, decentralized management of stormwater, rainwater harvesting, and use have been supported by policies and regulations that function as incentives for utilization by public and private actors [34].

Conversely, in the US, in states such as Colorado and Utah, policies banning rainwater and stormwater harvesting frustrate the ability to control runoff and water pollution. These are not sustainable and are barriers to cities dealing with scarce water resources [35].

On the other hand, still in the US, in the State of Pennsylvania, the stormwater management policy is consistent with other policies, such as the Clean Streams Act and other federal and state regulations; it incorporates these water-related policies and ensures that the demands of the involved parties have met [36].

Urban rainwater harvesting policies can support a dual-mode water supply system in Singapore [37], and in Indian megacities, a model, involving rainwater harvesting and efficiency improvement, concludes that these two policies combined can satisfy the criteria of efficiency, reliability, equity, financial viability, and revenue generation [38]. In South

Korea, rainwater harvesting is becoming more important, and governments have published a series of utilization policies and regulations [39].

As they are more cost-effective and compatible with the urban water cycle, several policies involving source pollution control techniques (e.g., retention, infiltration, and reuse) have been implemented in Europe (the UK, France, Sweden, Denmark, Germany, the Netherlands, and Greece) for the management of urban stormwater [40].

### 3.3.3. Theme 3: Regulation

There are two sides to the issue of regulation: the first addresses the regulation of the quality-of-service provision, basically linked to technological issues and the techniques and practices applied in the management of stormwater systems, closely linked to issues of pollution and flooding, and the second addresses economic regulation, linked to issues involving forms of financing and economic sustainability of actions and systems.

Most of the articles deal with quality regulation and a small number with economic regulation, often in a superficial way, with little questioning of the incentives for the actors to participate in the financing and investment and, even less, present alternatives for its realization. This may be due to the lack of institutional and political structuring of stormwater management in many cities, among other factors. The costs of designing and building sustainable drainage infrastructure may be easily known, but the lifetime costs, including operation and maintenance, are not. The life expectancy of these projects is uncertain and so are the operation and maintenance costs. The lack of knowledge about the performance and benefits to be obtained over their lifetime makes the risk of investment high for both the public and private sectors [10]. This may be the reason for the scarcity of articles about urban drainage economic regulation.

In urban stormwater management, the lack of data on system costs and performance is an obstacle to all kinds of initiatives, including economic regulation. This absence becomes an impediment to projects that can be financed, whether new or even retrofit (which hardly exist today), in addition, there is a lack of development of tools that allow the application of private resources [32]. Despite this reality, when the limits of regulation are reached or are insufficient, market-based tools can be attractive to encourage private owners to install GI, as shown in Table 9. In the US, in 400 cities, several pioneering initiatives impose tariffs on impermeable areas, but 70 cities in 20 states offer incentives through credits based on these tariffs, for the installation of GI [32].

There are also initiatives, such as that of the US Congress, that proposed an addendum to the Clean Water Act (CWA), to charge the federal government for stormwater management fees, where applicable, and proposals to finance social tariffs for those who cannot afford the fees. In the US, public health and environmental regulations make stormwater management an important issue and simultaneously a resource for water supply, especially in semiarid regions [41]. The decentralization of stormwater management is a strategy that seeks to reduce the flows and optimization of existing systems and those to be built. In the first case, there is difficulty in implementing decentralization without the participation of existing owners, and, thus, regulation should incorporate strategies of involvement and incentives for participation using institutionalized instruments, e.g., compulsory connection and tariffs [13]. Investments in stormwater GI are increasing in US cities and have the potential to carry out stormwater management along with other services, such as, e.g., ecosystems dealing with issues of urban heat islands and urban environments, and simultaneously bringing economic and regulatory changes [26], but its implementation remains slow [32,42].

**Table 9.** Potential market or quasi-market mechanisms and examples (Source: [32], adapted).

| Policy Mechanisms | Example Applications |
|---|---|
| Stormwater fees and discounts: A fee on runoff quantity or impervious area is enforced and discounts for installing GI are provided. | Seattle: enforces annual flat fee for single-family and duplexes smaller than 929 m². For all other cases, the annual fee is based on impervious areas. Portland: enforces off-site (65%) and in-site (35%) charges separately. Flat rate for a single family to 4 plex residences. Other cases: rates per 92.9 m² of impervious area. The Clean River Rewards Program: provides up to 100% of discounts for the on-site portion of the charge. |
| Allowance market: Tradable allowances of discharge are distributed among landowners, who are required to manage additional quantities. Those who can manage more than required can sell their allowances to others. | Washington DC: Stormwater Retention Credit (SRC) Program—landowners obtain SRC's credits for voluntary reduction of stormwater runoff (one SRC credit per additional gallon reduced above required reduction) using GI. The owners can bank for future use or trade their SRCs in an open market when others are willing to buy and use regulatory requirements for retaining stormwater [43]. It's the first kind of use of this mechanism in the US. |
| Payment for ecosystem services: Owners are paid for providing ecosystem services such as flood mitigation, carbon storage, and water purification. | US cities (New York, NY; Syracuse, NY; Boston, MA; Portland, OR; Seattle, WA): for the protection of watersheds that are their critical sources of water supply (Mercer et al., 2011). Countries (China, South Africa, Mexico, Costa Rica, Nicaragua, and also in the US): for many other ecosystem services [44]. |
| Rebates, credits, and installation financing: Tax credits, financing, and reimbursements to landowners who install GI. | Philadelphia's Tree Vitalization Rebate Program: provides a $25 rebate for planting a tree. The Rain Check Program: provides rain barrels for free and/or helps construct downspout planters, rain gardens, or porous paving for a reduced price. Seattle: City and King County pays to pay up to the total costs of rain gardens and cisterns. |
| Development incentives: Benefits to developers including expedited permitting, bonuses for floor area, height, density, and space. | Chicago: offers expedited permitting process for projects meeting Leadership in Energy and Environmental Design (LEED) criteria. Philadelphia: Provides floor area ratio and height bonuses (up to 400% and 10.97 m, respectively) for installing green roofs. Portland: provides floor area bonus up to 300% of the eco-roof area installed. |
| Grants and awards: Money is given directly to individual landowners or communities for installing GI. | Chicago's Green Roof Program: provides $5,000 to residential and small (<929 m²) commercial buildings. Portland's Community Watershed Stewardship Program: provides up to $10,000 for watershed restoration activities. Philadelphia's Stormwater Management Incentive Program (SMIP): provides grants for qualified non-residential owners. Green Acres Retrofit Program (GARP): provides grants for qualified contractors, companies, or project aggregators. |

In Israel, the transition from conventional to SUDS management brought changes to the regulation system, particularly incorporating infiltration aspects within the socio-institutional framework [45].

In Bangkok, Thailand, limitations on the regulatory framework were reported, together with financial and physical barriers, weak institutional frameworks, and poor community involvement in the implementation of stormwater source control and catchment approaches [46].

The scarcity of water in urban areas necessitates consideration of the use of rainwater, but for this, it is necessary to regulate and encourage its use, especially to face the risks involved [47].

## 4. Discussion

The literature review carried out in this study, based on a survey of articles published over the last decades, since 1980, aimed at verifying what is present (or absent) in these

documents, to enable an understanding of the subject of urban stormwater management beyond the technological challenges, involving issues that relate it to PIR.

The intended contribution was to systematize the existing knowledge in the accessed literature and, using electronic resources, e.g., machine learning, to present the themes, and the existing relations between them, that allow one to have an idea of what has been and what is being studied about urban stormwater management. In the literature, there are papers with interesting and comprehensive discussions and reflections, especially about those themes more linked to the traditional view of stormwater management and drainage and less linked to the sustainable management of stormwater. However, part of the literature focuses on the new paradigm shift, that is, stormwater as a resource and its management and control at the source, place of precipitation, when possible, and the search for a way to make its management economically viable, using regulation.

The results show that there are gaps in the understanding of the contribution of the tripod, represented by the PIR, to the financing agenda needed, for example, to achieve the goals of SDG6, sanitation for all. Although financing alone is a fundamental aspect, it is not enough to meet the needs that exist for planning, executing and monitoring the capacity of the institutions that will have to manage the funds obtained to finance infrastructure projects, their operation and maintenance, as well as social and environmental demands.

The instruments found in the literature, that exist today, and many of them in the USA, Australia and China, for the latter especially from 2013 onwards, are still in their infancy, needing to mature to have their potential fully utilized, but they are promising, and are based on experience.

There is a significant increase in peer-reviewed papers referring to stormwater management system PIR's subjects, especially in the last decade (2010) and at the beginning of the 2020s, with 84 (75%) produced articles, which allows us to assume that there is a growing interest in stormwater management PIR subjects. Most are published by authors from academic institutions in the USA, 45 (40%), and Australia, 15 (13%).

The evidence shows that there are no articles dealing specifically with PIR, and when there are, the effects of PIR on the results are just superficially approached. Despite this, the three correlated themes are present in the narrative analysis, i.e., institutions and governance—organizational, institutional, and governance arrangements, policy implementation, and regulation.

A good example of the interconnections between the sides of the PIR tripod is the example, briefly presented in Table 8, referring to the RTLP, in which the multiplicity of objectives entails a growing number of actors, often with diverging interests, making visible the presence of PIR, classified, in this case, as external factors, while the actors are internal factors.

This study's attributions derive from the use of what exists in a single database, albeit ample (SCOPUS), from the natural exclusions by the choice and analysis software used and, even relative to the period in which it was researched (from 1980 until the 2020s), with the prevailing trends in each time and place, that is, in the time and space, of the research carried out.

The methodology has many risks of bias: keywords selected, papers that were not included, the automated content analysis (ACA), and frequency-based analysis can exclude some words and connections not highly visible to the automated software systems. The gaps show that the understanding of the PIR is fragile, as an important element for analyzing of results to be achieved, including SDG6, the financing and obtaining funds, guarantees and grants for urban stormwater services and infrastructure.

In addition to the effects on results, it matters to recognize not just the gaps in PIRs, but the so-called Good Enough Governance, or trying to focus not just on the gaps, but on the issues that give better results and in the "how" and "how long times" it will take to obtain results with the available resources in each country and context.

With the evolution of experience resulting from the implementation of new projects underway, such as the sponge cities, and others, especially based on the events and projec-

tions of climate change in each location, much will have to be produced in the literature on the subject, such as the publication "*Water Supply and Sanitation Policies, Institutions, and Regulation—Adapting to a Changing World*" [48].

## 5. Conclusions

This paper contributes to the literature because it provides an overview of the papers discussing PIR, related to stormwater management, so academia and practitioners can understand what exists, what are the gaps remaining on the subject and what is its importance for the paths toward the solutions required to best deliver services. The research process has obvious limitations, due to the sample, existing papers in the database, and by choice of words used in the search engine.

This research is successful because it allowed us to identify that this subject—PIR—is a reason for growing concern and reflection in all countries, showing its importance, but it was not possible to identify a unique approach that allowed the comparison of the effects and results of the PIR. This can be considered one of its major research challenges and limitations to the existing literature gap to be filled in future studies.

The lack of a single approach to such a complex subject should not surprise researchers, as the realities are quite diverse, which, however, does not prevent the search to identify a particular reason or specific incentive for all places, but principles that can guide incentives to the alignment of PIR, at least in most situations, without this meaning a search for the unintelligent standardization of general criteria.

This work shows that future research must focus on identifying PIR priorities and especially incentives for tripod alignment. Among the many now existing demands and goals, such as the demands posed by the Sixth Sustainable Development Goal (SDG6), by 2030—Ensure Water and Sanitation for All—the knowledge about how the PIR tripod alignment can provide contributions, not only to urban stormwater management, but to understand how to best deliver urban services, is a main subject, with an objective and focus to reduce poverty and increase people's wellbeing. Thus, this is the knowledge challenge posed to us.

**Author Contributions:** Conceptualization, C.N. and R.C.M.; methodology, C.N. and R.C.M.; software, C.N.; validation, C.N. and R.C.M.; formal analysis, C.N. and R.C.M.; investigation, C.N.; resources R.C.M.; data curation C.N and R.C.M.; writing-original draft preparation, C.N.; writing-review and editing, C.N. and R.C.M.; visualization C.N. and R.C.M.; supervision, R.C.M.; project administration, R.C.M.; funding acquisition R.C.M. All authors have read and agreed to the published version of the manuscript.

**Funding:** This research received no external funding.

**Data Availability Statement:** The data supporting results can be found in the text of the article.

**Conflicts of Interest:** The authors declare no conflicts of interest.

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
