# Peer review of "Policy, Institutions and Regulation in Stormwater Management: A Hybrid Literature Review"

_water, doi:10.3390/w16010186_

Round 1
Reviewer 1 Report
Comments and Suggestions for Authors
This work mainly studied the policies, management, and methods of stormwater. In the article, relevant papers were searched through search engines, and then the authors, countries, and institutions of the papers were analyzed, as well as the keywords in the papers. Finally, the above information is introduced in three themes. These three themes are Institutions and Governance, Policy Implementation, and Framework & Regulation. Through the introduction of these three themes, it was concluded that future research must focus on determining the priorities of PIR and especially the incentives aligned with the tripod. I gives these suggestions to the review.
(1)In “3.3.1. Theme 1”, I believe there are several areas that require modification. Firstly, the opening sentence, “There is a great concern, in terms of the sector organization, reflected in many articles, related to institutions and governance of the water systems”, should be supported by specific data such as the number of articles and the proportion of relevant articles. Next, When the author mentions stormwater, it is increasingly recognized as a resource rather than just a flood or pollution issue. However, this change also brings management risks and responsibilities. To ensure the accuracy of this viewpoint, the author should use relevant literature to confirm it.
Additionally, some terms in the article were not explained, such as what exactly is a sustainable drainage system and how it differs from traditional drainage systems in terms of advantages and disadvantages.
(2)In “3.3.2. Theme 2”,the sentence ‘”heme 2 is related to policy implementation and the framework” was the main theme of this section in Theme 2. However, it is unclear what this sentence aims to convey. Does it mean that ‘policy implementation and the framework’ were the key topics that scholars are emphasizing? If this is the case, then the examples given in this text may not be appropriate. Therefore, it is important to provide a clearer expression of the main theme.
(3)In “3.3.3. Theme 3”, the author presents a viewpoint regarding the lack of motivation for questioning actors to participate in financing and investment, as well as the absence of articles proposing alternative solutions for financing and investment. However, the paper does not provide any additional evidence to support this perspective. Therefore, the author should further provide relevant literature or data to substantiate the arguments made in the paper.
(4)In “conclusion 5”, some professional terms are not explained, which can lead to difficulties for readers to understand
Comments on the Quality of English Language
The English translation of this review still contains some unclear points and requires further improvement to enhance clarity of expression. We will continue to work on refining the English of this article to ensure that readers can understand its content more clearly.
Reviewer 2 Report
Comments and Suggestions for Authors
This is an interesting and useful overview paper. It uses a range of modern data searching techniques to find what literature exists on policies, institutions, and regulations (PIR) relating to stormwater, and finds a lack of consideration of all three together. There is quite a lot on each of those three areas, but not much good integration of them into a coherent picture of water policy (I am familiar mostly with California, USA, and can attest to the truth of this for our area and much of the western US). More interesting and significant is the concentration of research on stormwater PIR in the US and Australia. Focus in these two countries is understandable--most of Australia and all the southwestern USA are very dry, and when rain does come it is often intense and prolonged, leading to a famine-or-feast regime that forces management of stormwater, to drain it off when it falls but to save it when it appears. What is interesting is that other countries with the same problem, such as India, Ethiopia, and Iran, have little or no scholarship on this issue. To me, the most valuable part of this ms is the table of measures that are being taken, largely in the US. This brings together what we actually know and do. More on this would be very much appreciated.
The paper focuses also on the transition--at first slow, now rapid--from seeing stormwater purely as something to drain off, to seeing it as something we need to save. Already desperate water shortages in much of the world are being exacerbated by global climate change. Stormwater management must now take very serious account of the need to save all that is possible. The article could have done more with the "sponge cities" idea (mentioned a couple of times). There are even paving materials that blot up rainfall now.
This is definitely a preliminary paper--it is short and much of its length is taken up with methodology and introductory matter. The promise of highlighting the lacks and shortfalls of existing literature is not well enough fulfilled. We really need more on what is missing that could and should be done, e.g. in that area of PIR for conserving and saving water. Purifying stormwater is another issue not much covered here. Stormwater gets badly contaminated with everything from spilled fuels to fertilizers and pesticides. This presents a problem for using it.
The ms is certainly publishable as a preliminary look at the literature, and filling it out more would take time. (It could easily turn into a book.) There is a tradeoff here: this ms is really needed, and should be made available soon. Adding more would be very helpful, though, for anything that can be done quickly.
Comments on the Quality of English LanguageEnglish is good, but sometimes could be cut; much of the introductory material could be said more succintly. This is not a huge problem.
Reviewer 3 Report
Comments and Suggestions for Authors
The manuscript entitled “Policy, Institutions and Regulation in Stormwater Management: A Hybrid Literature Review” is submitted for possible publication in MDPI Water journal. The paper sets out to conduct a hybrid literature review of peer-reviewed papers in this field to provide an overview of how researchers have been studying policies, institutions, and regulation relations. Overall, I do not think the paper contains useful information.
Comments
1. The abstract lacks strong statements and outcomes. Authors ought to rewrite the abstract section. All standard elements of the abstract should be included.
2. Lines 22 to 52, this is very general and can be stated in 5 lines. I have a major concern here.
3. Section 3.3.1 – no proper flow of information or arguments. The same thing for sections 3.3.2 and 3.3.3.
4. The discussion section lacks depth and proper analysis. This section should be the most important.
5. The paper failed to identify gaps in the literature. Furthermore, no recommendations on how the different themes should be addressed.
6. Overall, I do not think the authors provided answers to the stated objectives. The contribution of the paper is below average.
7. Minor - The numbering of Tables and figures is rather strange and not in line with the journal’s layout.
Comments on the Quality of English LanguageExtensive editing of the English language required
Round 2
Reviewer 3 Report
Comments and Suggestions for Authors
I have no further comments on the revised version of the paper.
Comments on the Quality of English LanguageModerate editing of the English language required